# Rewiring Mood: Precision Psychobiotics as Adjunct or Stand-Alone Therapy in Depression Using Insights from 19 Randomized Controlled Trials in Adults

**DOI:** 10.3390/nu17122022

**Published:** 2025-06-17

**Authors:** Alexandra-Eleftheria Menni, Helen Theodorou, Georgios Tzikos, Ioannis M. Theodorou, Eleni Semertzidou, Veroniki Stelmach, Anne D. Shrewsbury, George Stavrou, Katerina Kotzampassi

**Affiliations:** 1Department of Surgery, Aristotle University of Thessaloniki, 54636 Thessaloniki, Greece; alexmenn@auth.gr (A.-E.M.); gtziko@auth.gr (G.T.); stelmachveroniki@gmail.com (V.S.); a_shrewsbury@yahoo.com (A.D.S.); 2Department of Sociology, School of Social Sciences, University of Crete, 74100 Rethymno, Greece; psy5057@psy.soc.uoc.gr; 3Hellenic Institute for the Study of Sepsis (HISS), 11528 Athens, Greece; itheodorou@med.uoa.gr; 4Library of AHEPA University Hospital, 54636 Thessaloniki, Greece; elenisemer@gmail.com; 5Department of Surgery, 417 NIMTS (Army Share Fund Hospital), 11521 Athens, Greece; stavgd@gmail.com

**Keywords:** probiotics, psychobiotics, add-on therapy, depression, major depression disorder, subthreshold depression

## Abstract

**Background:** Depression is a leading contributor to global disability, with a large proportion of patients showing inadequate responses to conventional antidepressants. Probiotic bacteria with psychotropic potential seem to be an emerging treatment option, either alone or in conjunction with depression symptom management. **Objective:** To critically review the Randomized Clinical Trials (RCTs) whose primary focus was to evaluate the efficacy of probiotics/psychobiotics to ameliorate depression status, quantified via validated psychometric tools. **Methods:** A comprehensive literature search of the PubMed and Scopus databases (January 2014–January 2025) was conducted to identify RCTs with the primary aim of improving depression status in adults taking probiotics in comparison to those taking a well-defined placebo. **Results:** Nineteen RCTs met the inclusion criteria, with all demonstrating a significant amelioration of depression status after probiotic/psychobiotic treatment, taken either as a stand-alone treatment [*n* = 5] or as an adjunctive treatment to antidepressant therapy [*n* = 10]. However, only in 14 studies was a significant improvement achieved at the end of treatment over a placebo, which also showed an improvement against the baseline. In total, 7 out of 10 studies with probiotics as an add-on therapy and 7 out of the 9 with probiotics, either as a monotherapy or with a different percentage also taking antidepressants, exhibited a significant amelioration of depression status against placebo treatment. **Conclusions:** Probiotics, particularly multi-strain preparations and certain well-characterized single strains, seem to be noticeably beneficial in alleviating depressive symptoms in adults. However, there is an urgent need for large-scale randomized clinical trials with well-defined specific psychobiotic strains in order to confirm the most effective strains.

## 1. Introduction

Depression is one of the most prevalent and burdensome psychiatric disorders and one of the most significant challenges faced by worldwide health services [1,2]. According to the World Health Organization (WHO), its global incidence is approximately 4.4% or over 300 million people worldwide [3], while others estimated a lifetime prevalence of approximately 16–20% [4], with up to 10% of the world’s elderly population experiencing depression [5]. It is characterized by persistent low moods, anhedonia, a loss of energy, a decreased interest in daily activities, a decreased appetite, sleep disorders, and a decreased thinking ability and cognitive deficits [1,4,6,7,8], with symptoms significantly affecting quality of life and often leading to suicide [5,9,10].

According to the guidelines issued by the National Institute for Health and Care Excellence, the first-line treatment for depression is based on monoamine-related pathways with or without psychotherapy [11]; however, the rate of ineffective therapies is high: almost two-thirds of patients experience some degree of nonresponse to first-line treatments [12,13], while another one-third may have treatment-resistant depression [14].

Over the years, the pathogenesis of depression has been connected with the disruption of multiple metabolic pathways, including neurotransmission, and oxidative stress [15,16,17], while the microbiome–gut–brain axis, which involves the immune, neural, endocrine, and metabolic pathways between the gut and the brain, has gained the spotlight [18,19,20,21,22]. On the other hand, a substantial body of work indicates that psychiatric disorders are clearly related to alterations in the gut microbiome composition, since individuals with depression were found to exhibit a decline in anti-inflammatory butyrate-producing bacteria, such as *Faecalibacterium*, and an increased abundance of pro-inflammatory microbial genera [19,23,24,25,26], implicated, among others things, in the modulation of tryptophan metabolism [27].

On this basis, it has been suggested that the modification of the gut microbiota to produce benefits for brain function may have translational applications in the modulation or even treatment of neuropsychiatric illnesses [28,29,30]. From this point of view, *psychobiotics*, a specified group of probiotics, the name of which stands for probiotics that support mental health benefits in patients with psychiatric illness, have been one of the most discussed exogenous interventions since their introduction by Ted Dyan and John Cryan in 2013 [31,32,33]. They have the unique characteristics of producing and delivering neuroactive substances, such as gamma-aminobutyric acid and tryptophan, which assist in serotonin synthesis in the host [34,35], act on the brain–gut axis [36,37], and generally manifest beneficial effects on depression, anxiety, and other psychopathological disorders by means of modifying the disturbed microbiome abundance [19,38,39,40,41,42,43,44,45,46].

In a 2021 meta-analysis of seven RCTs—404 patients—the authors [23] referred to different probiotic species alleviating depressive symptoms when administered adjunctively to antidepressants. However, there are multiple other RCTs demonstrating either a beneficial effect or reporting promising but inconclusive outcomes or even poor or conflicting results; the findings are partially affected by the study population as well as the small or un-homogeneous sample sizes [24,28,40,42,43,47,48,49,50,51,52,53,54,55,56,57,58,59,60]. A typical example is a meta-analysis of 19 double-blinded, randomized, placebo-controlled trials, published between 2010 and 2019, on the effect of probiotics on depressive symptoms [45,47]; it is notable that only 3 of them included individuals with Major Depression Disorders [MDD] [42,61,62], while the remaining 16 studies were conducted with controls without depression or other clinical populations, e.g., irritable bowel syndrome, diabetes with coronary heart disease, and fibromyalgia [43,45,47,63,64].

On the other hand, it has been widely thought—based on various systematic reviews and meta-analyses—that the individualized effects of a probiotic formula are both absolutely strain-dependent and strain-specific, and that a regime of collaborating microorganisms might be more beneficial than a single strain [65]. In this vein, in a meta-analysis of probiotic treatment in clinical populations of major and co-morbid depression it is acknowledged that multi-species probiotics, consisting of both *Bifidobacteria* and *Lactobacilli* strains, work better than single strains in mood improvement [48]. Therefore, the development of a novel and more efficient treatment approach is urgently needed.

Thus, the aim of the present review was to search and analyze the existing literature on depression treatment with probiotics/psychobiotics alone or as a co-treatment with the antidepressants participants already take, with the strict inclusion criteria being (i) depression as the primary diagnosis; (ii) the diagnosis established using at least one psychometric test; (iii) depression improvement as the clear primary objective of study; (iv) the alleviation of the symptoms documented by the same psychometric tests with which the diagnosis was originally established; (v) only randomized controlled trials; (vi) the ingredients of the placebo treatment being precisely defined.

## 2. Materials and Methods

### 2.1. Search Strategies

A comprehensive literature search was performed between 5 and 15 January 2025 in the PubMed and Scopus databases using a variety of different search terms in order to find all relevant articles. The following search algorithm was employed: [“depression” OR “MDD” OR “major depression episodes”] AND [“probiotics” OR “psychobiotics”]. Then, filters for (i) time; a timeframe from January 2014 to January 2025 was used, since the year 2014 signaled the beginning of publications on “psychobiotics”, following the adoption of the term by Ted Dinan and John Cryan at the end of 2013 [32]; (ii) study design; searching for “randomized clinical trials” OR “clinical trial(s)” OR “RCTs”; (iii) age; “adults only”, were applied, without any language restriction. Additionally, the reference lists of four more recent review articles, all published in 2024, were manually scrutinized by the same reviewers in order to identify possible missed relevant studies.

### 2.2. Eligibility Criteria

Two authors, working independently, went through the full texts of all selected publications to search manually and reject those articles not fulfilling the inclusion criteria; in the case of discrepancy or disagreement, a third author was involved to resolve the issue.

The inclusion criteria were (i) depression, in adults only, to be the primary diagnosis, established by at least one, well-validated, psychometric test; (ii) depression improvement to be clearly defined as the primary objective of the study; (iii) the symptom alleviation to be based on the same psychometric test(s) as the diagnosis of depression; (iv) only randomized controlled trials to be included; (v) the ingredients of the placebo [comparator] treatment to be precisely defined.

### 2.3. Data Extraction

Data extraction was then performed by the same authors, through recording the following items: article citation, year of publication, sample size, age of participants, depression status, psychometric test(s) used for the diagnosis of depression, co-morbidity related to depression, antidepressant treatment, probiotic(s) used, daily dose, treatment duration, whether live bacteria or postbiotics (defined as “a preparation of inanimate microorganisms and/or their components that confers a health benefit on the host” [66]) were used, prebiotics as co-treatment, other co-treatments related to depression, positive results (meaning a significant reduction in the psychometric test(s) score), and negative results, (meaning an almost similar reduction in the psychometric test(s) score in both treatment and placebo groups).

## 3. Results

Our search strategy in the PubMed and Scopus databases initially identified a total of 1169 records. After setting the timeframe, the number of articles was reduced to 1072, from which 103 were initially found to be RCTs in adults. Then, manually selected by title/abstract, another 61 were excluded, leaving 42 RCTs. After full-text reading, 19 articles remained for analysis. From these 19 studies, 14 included depressed but otherwise healthy subjects, while of the remaining 5 studies, all the participants had the same co-morbidities: irritable bowel syndrome in 2 studies, and coronary artery disease, abdominal obesity/metabolic syndrome, and gastrointestinal cancer surgery patients on a chemotherapy course in 1 each. The search outcome is displayed as a flowchart in Figure 1.

### 3.1. Methodological Considerations

Although this analysis does not follow a systematic review framework, a basic quality appraisal of the included RCTs was conducted, based on commonly used methodological criteria, including, among others, the use of randomization, blinding, sample size, clearly defined outcomes, and whether a placebo control was used. All of the studies provided sufficient information regarding randomization, blinding, the selection of participants, and validated psychometric tests used for both depression diagnosis and severity quantification and clearly defined outcomes. In general, the majority of RCTs demonstrated a high standard of design, with adequate methodological rigor and clear reporting, thus leading to the increased reliability of their findings regarding the effects of probiotics and related interventions on depressive symptoms.

However, there were some limitations: (i) The sample sizes were generally modest, which, in some cases, limits the statistical power, with the number of participants being 50 or fewer, in seven studies, between 51 and 100 in seven studies, and over 100 in the other five studies. (ii) The methodology of whether probiotics were used as an add-on therapy, stand-alone therapy, or other method make the study population heterogenous, limiting the similarity in response to treatment. In this review, there are four studies in which there was no restriction on participants’ entrance either receiving or not receiving antidepressants, so it is not clear whether the results are due to antidepressant pharmacotherapy or not. Furthermore, according to common sense, individuals with depression needing pharmacological support are not comparable to those having no need. (iii) The psychometric tools used in each study were not similar—19 studies used 11 different psychometric tests to document depression. However, since the final purpose of each study was the significant (quantitative—whatever the scale used] amelioration of depression status in relation to a placebo, there is no problem (Appendix A) is inserted depicting the differences between tools). (iv) Only a few studies implemented intention-to-treat analyses or reported dropout rates in detail. (v) Lastly, while some trials clearly defined strain composition, others lacked transparency in formulation specifics, impeding reproducibility and cross comparison.

### 3.2. Overview of Studies (Table 1 and Table 2)

One of the earlier studies was that of Akkasheh, G. et al. [42]. They presented their positive findings on the clinical symptoms of depression and metabolic health markers in 40 individuals with MDD, with the diagnosis based on Diagnostic and Statistical Manual of Mental Disorders IV [DSM-IV] criteria, randomized to receive a probiotics formula containing *Lactobacillus acidophilus*, *Lactobacillus casei*, and *Bifidobacterium bifidum* species, in a concentration of 2 × 10^9^ cfu/capsule/day or an identical placebo, for 8 weeks. The Beck Depression Inventory score (BDI)—as a primary outcome measure—was used at baseline for the assessment of depression severity, as well as at study conclusion at 8 weeks. Probiotics were found to significantly improve depression status by −5.7 ± 6.4 units of the BDI total score, in relation to the placebo treatment, [−1.5 ± 4.8, *p* = 0.001] [42].

The following year, the postbiotic *Bifidobacterium longum* NCC3001 was tested on its ability to induce a reduction of at least 2 points on the Hospital Anxiety and Depression Scale [HAD] for depression and/or anxiety scores [67]. The study was conducted in a cohort of 44 adults with irritable bowel syndrome, experiencing either diarrhea or a mixed stool pattern, as per the Rome III criteria, and experiencing, as a co-morbidity, depression and/or anxiety [HAD scores between 8 and 14]. The participants were randomized to receive either *B. longum* NCC3001 [1 × 10^−10^ cfu/gr] or an identical placebo for 6 weeks. At 6 weeks, a significantly greater percentage [64% vs. 32%, *p* = 0.04] of *B. longum*-treated individuals experienced a ≥2 point reduction in depression score in comparison to the placebo, with the result being directly related to an adequate relief of IBS symptomatology [67].

Ghorbani et al. from Iran [61] gave two capsules of a synbiotic formula (Familact®, Zist Takhmir Co, Tehran, Iran) containing *L. casei* [3 × 10^8^ cfu/gr], *L. acidophilus* [2 × 10^8^ cfu/gr], *L. bulgaricus* [2 × 10^9^ cfu/gr], *L. rhamnosus* [3 × 10^8^ cfu/gr], *B. breve* [2 × 10^8^ cfu/gr], *B. longum* [1 × 10^9^ cfu/gr], and *Streptococcus thermophilus* [3 × 10^8^ cfu/gr] plus 100 mg fructo-oligo-saccharide [FOS] as a prebiotic or the identical placebo along with 20 mg Fluoxetine daily to 40 adults with moderate depression.

**Table 1 nutrients-17-02022-t001:** Fourteen studies which exhibited positive results regarding depression status after probiotic administration.

	Authors, Year, [Ref]	Country	N	Depression Status	Diagnosis Based on	Co-Morbidity	Duration [Weeks]	Antidepressants	Probiotic Strains	Co-Treatment	Dose [cfu/d]
1.	Akkasheh, G. et al., 2016 [42]	Iran	40	MDD	DSM-IV	_	8	not reported	*L. acidophilus*, *L. casei*, *B. bifidum*		6 × 10^9^
2.	Pinto-Sanchez, M.I. et al., 2017 [67]	Switzerland	44	D	HAD	IBS	6	excluded	*B. longum* NCC3001		1 × 10^10^
3.	Ghorbani, Z. et al., 2018 [61]	Iran	40	mD	DSM-V	_	6	add-on	**Familact^®^:** *L. casei*, *L. acidophilus*, *L. bulgaricus*, *L. rhamnosus*, *B. breve*, *B. longum*, *S. thermophilus*	FOS	8.6 × 10^9^
4.	Chahwan, B. et al., 2019 [55]	Australia Netherlands	71	D	BDI-II	_	8	excluded	**Ecologic^®^ Barrier:** *B. bifidum W23*, *B. lactis W51*, *B. lactis W52*, *L. acidophilus W37*, *L. brevis W63*, *L. casei W56*, *L. salivarius W24*, *Lc. lactis W19*, *Lc. lactis W58*		1 × 10^10^
5.	Kazemi, A et al., 2019 [62]	Iran	110	MDD	BDI	_	8	add-on	*L. helveticus R0052*, *B. longum R0175 [CNCM strain I-3470]*		1 × 10^10^
6.	Saccarello, A et al., 2020 [68]	Italy	90	D	Z-SDS	_	6	add-on	*L. plantarum HEAL9*	SAMe	1 × 10^9^
7.	Schaub, A.C. et al., 2022 [53]	Switzerland	60	MDD	HAM-D	_	4	add-on	**Vivomixx^®^:** *Streptococcus thermophilus NCIMB 30438*, *B. breve NCIMB 30441*, *B. longum NCIMB 30435*, *B. infantis NCIMB 30436*, *L. acidophilus NCIMB 30442*, *L. plantarum NCIMB 30437*, *L. paracasei NCIMB 30439*, *L. delbrueckii* ssp. *Bulgaricus NCIMB 30440*		9 × 10^11^
8.	Tian, P. et al., 2022 [27]	China	45	MDD	HDRS-24, MADRS,	_	4	no restriction	*B. breve CCFM1025*		1 × 10^10^
9.	Ullah, H. et al., 2022 [69]	Italy	80	SD	PHQ-,	_	38	add-on	*L. helveticus Rosell^®^-52*, *B. longum Rosell^®^-175*	SAMe	3 × 10^9^
10.	Moludi, J. et al., 2022 [70]	Iran	96	D	BDI-II	CAD	8	excluded	*L. rhamnosus GG*		1.9 × 10^9^
11.	Nikolova, V.L. et al., 202, [54]	Poland	50	MDD	HAMD-17	_	8	add-on	**BioKult Advanced^®^:** *L. paracasei PXN37*, *L. plantarum PXN47*, *L. rhamnosus PXN54*, *B. subtilis PXN21*, *B. bifidum PXN23*, *B. breve PXN25*, *B. longum PXN30*, *L. helveticus PXN35*, *L. lactis* ssp. *lactis PXN63*, *S. thermophilus PXN66*, *B. infantis PXN27*, *L. delbruecklii* ssp. *bulgaricus PXN39*, *L. helveticus PXN45*, *L. salivarius PXN57*		8 × 10^9^
12.	Strodl, E. et al., 2024 [71]	Australia	120	MDD	DSM-V, BDI-II	_	8	no restriction	**NRGBiotic^®^:** *L. acidophilus*, *B. bifidum*, *S. thermophilus*		8 × 10^10^
13.	Tzikos, G. et al., 2025 [72]	Greece	266	D	BDI-II, HDRS-17	SOP	4	excluded	*B. animalis* subsp. *lactis LMG P-21384*, *B. breve DSM 16604*, *B. longum DSM 16603*, *L. rhamnosus ATCC 53103*		1.76 × 10^11^
14.	Elahinejad, V. et al., 2025 [73]	Iran	50	MDD	DSM-IV, HDRS-24	_	8	add-on	*L. helveticus Rosell^®^-52*, *B. longum Rosell^®^-175*		6 × 10^9^

IBS; Irritable Bowel Syndrome. CAD; Coronary Artery Disease. SOP; Surgical Oncology Patients. D; Depression. MDD; Major Depressive Disorder. SD; Subthreshold Depression. mD; moderate Depression.

**Table 2 nutrients-17-02022-t002:** Five studies which exhibited negative results regarding depression status after probiotic administration.

	Authors, Year, [Ref]	Country	N	Depression Status	Diagnosis Based on	Co-Morbidity	Duration [Weeks]	Antidepressants	Probiotic Strains	Co-Treatment	Dose [cfu/d]
1.	Rudzki, L. et al., 2019 [59]	Scotland, UK, and Poland	60	MDD	DSM-IV-R	_	8	add-on	*Lactobacillus Plantarum 299v* (*LP299v*)	-	2 × 10^10^
2.	Reininghaus EZ et al., 2020 [33]	Austria	82	D	MINI	_	4	add-on	**OMNi-BiOTiC^®^ Stress Repair:** *B. bifidum W23*, *B. lactis W51*, *B. lactis W52*, *L. acidophilus W22*, *L. casei W56*, *L. paracasei W20*, *L. plantarum W62*, *L. salivarius W24, L. lactis W19*	Biotin	7.5 × 10^9^
3.	Gawlik-Kotel-nicka, O. et al., 2024 [2]	Poland	116	D	MADRS	AO/MeS	8	no restriction	*L. helveticus Rosell^®^-52*, *B. longum Rosell^®^-175*	-	3 × 10^9^
4.	Sarkawi, M. et al., 2024 [74]	Malaysia	124	SD	CESD-R	IBS	12	excluded	*Lactobacillus acidophilus LA-5*, *Lactobacillus paracasei L. CASEI-01*	-	2 × 10^9^
5.	Lin, S.K. et al., 2024 [75]	Malaysia	32	MDD	DSM-V, HAMD-17, DSSS		8	add-on	*L. plantarum PS128*	-	3 × 10^10^

In another study, the Hamilton Rating Scale for Depression [HAM-D] was measured and assigned a score from 17 to 23 for 6 weeks. The antidepressant treatment had been previously given for 4 weeks as pre-treatment. A synbiotic plus Fluoxetine was found to significantly decrease the HAM-D score in comparison to the placebo plus Fluoxetine group, with the mean [SD] reduction from baseline to the end of treatment being −19.25 [1.71] vs. −17.75 [2.05] in placebo-treated cases, *p* = 0.013 [61].

Seventy-one mildly [score 12–28] or severely depressed [score > 28] participants according to the BDI-II not taking any antidepressants were randomly allocated to a postbiotic mixture or placebo for 8 weeks, aiming to verify the positive effect of treatment in alleviating depression symptoms [55]. The commercially available Winclove’s Ecologic^®^ Barrier (Winclove probiotics, Amsterdam, The Netherlands) containing nine bacterial strains, *Bifidobacterium bifidum W23*, *Bifidobacterium lactis W51*, *Bifidobacterium lactis W52*, *L. acidophilus W37*, *Lactobacillus brevis W63*, *Lactobacillus casei W56*, *Lactobacillus salivarius W24*, *Lactococcus lactis W19, and Lactococcus lactis W58,* was taken in a total dose of 1 × 10^10^ cfu/day. There was no significant improvement in the probiotic group regarding the BDI-II score, as well as the Depression Anxiety Stress Scale-21 items [DASS-21] and the Beck Anxiety Inventory [BAI] when the participants were assessed all together. However, when assessed separately, based on the initial depression severity, a statistically significant difference was revealed only in the severely depressed group and only for the BDI-II score, in relation to the placebo. However, we should mention that there was a high attrition rate of 34%, that is, 47 participants [55].

Rudzki et al. [59] investigated the influence of *Lactiplantibacillus plantarum 299v* [10 × 10^9^ cfu/capsule, twice daily for 8 weeks] on the intensification of SSRI antidepressant treatment in 60 major depression disorder participants, against a placebo. Unfortunately, the treatment in this cohort failed to result in an amelioration of depression severity as assessed by the HAM-D 17 compared with a placebo, although a decrease in kynurenine concentration and an increase in 3HKYN:KYN ratio was prominent in the *L. plantarum* group [55].

One hundred and ten individuals with depression were randomized to receive the combination of postbiotics *Lactobacillus helveticus R0052* and *Bifidobacterium longum R0175* [*CNCM strain I-3470*] at a concentration of 10 × 10^9^ cfu/sachet, the prebiotic galacto-oligosaccharide, or the identical placebo for 8 weeks, with all individuals being under antidepressant treatment for at least 3 months before entering the protocol [62]. From baseline to week 8, the postbiotics regime—but not the prebiotic—resulted in a significant decrease in BDI score in comparison to the placebo [62].

Reininghaus et al. [33] tested the commercially available regime “OMNi-BiOTiC^®^ Stress Repair” (Allergosan, Graz, Austria) administered for 28 days against an identical placebo in reducing depression symptoms in 82 individuals with depression, with their diagnosis being established with the Mini International Neuropsychiatric Interview [MINI] and the individuals already being under a standard antidepressive treatment. The probiotic formula contained *B. bifidum* W23, *B. lactis* W51, *B. lactis* W52, *L. acidophilus* W22, *L. casei* W56, *L. paracasei* W20, *L. plantarum* W62, *L. salivarius* W24, and *L. lactis* W19, at a concentration of 7.5 × 10^9^ cfu/day; both groups were also taking 125 mg D-Biotin [vitamin B7], 30 mg of common horsetail, 30 mg of fish collagen, and 30 mg of keratin plus matrix. Both groups were found to have exhibited a significant improvement in their depression symptoms, and there was no significant difference in the decrease between the probiotics and the placebo group in any of the psychiatric tests used, that is the HAM-D, the BDI-II, and the Symptom Checklist-90-Revised form [SCL-90] [33].

S-Adenosylmethionine [SAMe] is one of the naturally found compounds recommended by the Canadian Network for Mood and Anxiety Treatments (2016) as a first- or second-line “natural” treatment alternative to conventional antidepressants for reducing the overall symptomatology of mild-to-moderate depressive disorders. Saccarello A et al. [68] assessed the effects of a 6-week treatment with *Lactobacillus plantarum HEAL9* [1 × 10^9^ cfu/day] plus 200 mg of SAMe or an identical placebo in 90 patients with depression having a total Zung Self-Rating Depression Scale [Z-SDS] score from 41 to 55. A significant improvement in depression status was clear from week 2 and thereafter regarding the Z-SDS total score, as well as in the core depression subdomain in week 6 [68].

A short-term but high-dose probiotic supplementation was used in 60 inpatients of a psychiatric clinic, who were under a standard antidepressant treatment for mild depression [score > 7] according to HAM-D. The participants were randomized into using a commercially available probiotic formula Vivomixx® (Mendes SA, Lugano, Switzerland) containing eight different strains [*Streptococcus thermophilus NCIMB 30438*, *Bifidobacterium breve NCIMB 30441*, *Bifidobacterium longum NCIMB 30435* (re-classified as *B. lactis*), *Bifidobacterium infantis NCIMB 30436* (re-classified as *B. lactis*), *Lactobacillus acidophilus NCIMB 30442*, *Lactobacillus plantarum NCIMB 30437*, *Lactobacillus paracasei NCIMB 30439*, and *Lactobacillus delbrueckii* subsp. *Bulgaricus NCIMB 30440* (re-classified as *L. helveticus*)] at a daily dose of 9 × 10^11^ cfu/day or an identical placebo for 4 weeks. The probiotic treatment was found to significantly decrease the HAM-D score over time in comparison to the placebo [53].

The psychotropic potential of *Bifidobacterium breve CCFM1025* was assessed by Tian, P. et al. [27] in 45 individuals with MDD [Hamilton Depression Rating Scale—24 item [HDRS-24] score over 14]. Before randomization to receive either the psychobiotic *B. breve CCFM1025* in a dose of 10^10^ CFU/g [*n* = 20] or maltodextrin as the placebo [*n* = 25] for 4 weeks, the participants were additionally assessed using the HDRS-24, the Montgomery–Asberg Depression Rating Scale [MADRS], the Brief Psychiatric Rating Scale [BPRS], and the Gastrointestinal Symptom Rating Scale [GSRS]. At the study’s conclusion, the probiotic-treated group revealed a significant decrease in all studied psychometric scores with a parallel improvement of GSRS in relation to the placebo-treated group [27].

A major problem in depression treatments is patients’ non-response, which frequently occurs in the case of subthreshold depression, as antidepressants may yield negative effects surpassing their benefits, thus resulting in the cessation of treatment by the patient. Ullah et al. [69] investigated the effectiveness of a dietary supplement in reducing depression symptoms; the supplement contained the known psychobiotics *Lactobacillus helveticus Rosell^®^-52* [2.7 × 10^9^ cfu/day] and *Bifidobacterium longum Rosell^®^-175* [0.3 × 10^9^ cfu/day], along with S-adenosyl methionine [SAMe, 200 mg/day], magnesium oxide [93.30 mg/day], and Vitamin B6 [1.70 mg/day], which is classically given to individuals diagnosed with subthreshold depression who are do not qualify for standard medication. Eighty patients with a mild [less than 9] or moderate [less than 15] score, according to Patient Health Questionnaire-9 [PHQ-9], and additionally assessed with the HAM-D questionnaires, were randomly recruited into 32 weeks of treatment using the above-mentioned compound or placebo, followed by a 6-week follow-up. In particular, all subjects, after a 4-week run-in phase receiving the placebo [t1], entered the 12-week clinical trial phase [dietary supplement or placebo] [t2], then a 4-week wash-out period [no supplement—no placebo] [t3], followed by another 12-week cross-over clinical trial of placebo or food supplements [t4], and finally a 6-week follow-up period [t5]. The individuals taking the dietary supplement exhibited a significant decrease in HAM-D score as a result of the alleviation of depression symptoms in relation to the placebo, with the reduction being highly related to the sequence of the experimental treatments, meaning that participants receiving the placebo first [between t1–t2] followed by the dietary supplement [between t3–t4] had a more improved depression status. This study, having the major strength of the longer treatment duration compared with the other clinical trials, supports the finding that this dietary add-on psychobiotic improves depression status and enhances the quality of life in patients ineligible for antidepressant treatments; however, there is no data for the group taking probiotics only without the add-on treatment with magnesium, SAMe, and Vitamin B6 [69].

Ninety-six individuals with depression with a background of coronary artery disease were randomized to two months of treatment of either *Lactobacillus Rhamnosus* GG [1.9 × 10^9^ cfu/day] or a placebo, with or without inulin as a prebiotic in a dose of 15 gr/day—four groups in total [70]. The aim was the evaluation of the effect of the probiotic alone or in combination with inulin in the amelioration of depression status, as assessed by BDI-II, in relation to the placebo [maltrodextrin], with identical packaging for both the probiotic [capsule] and prebiotic [sachet]. Both of the treatment groups of either L. rhamnosus alone or inulin alone had significantly reduced depression and anxiety scores, as assessed by BDI-II and the State-Trait Anxiety Inventory [STAI]; however, the co-administration was found to amplify the improvement of the psychological outcomes in relation to the corresponding placebo groups [70].

A 14-strain probiotic regime (BioKult Advanced®,ADM Protexin, UK), was tested for the estimated size of the intervention effect of probiotics as an 8-week adjunctive treatment for 50 outpatients with a primary diagnosis of MDD [HAMD-17 greater than 13], who were taking a selective serotonin reuptake inhibitor at a stable dose for 6 or more weeks but having an incomplete response [54]. The participants who were administered the probiotics *L. paracasei PXN37*, *L. plantarum PXN47*, *L. rhamnosus PXN54*, *B. subtilis PXN21*, *B. bifidum PXN23*, *B. breve PXN25*, *B. longum PXN30*, *L. helveticus PXN35*, *L. lactis* ssp. *lactis PXN63*, *S. thermophilus PXN66*, *B. infantis PXN27*, *L. delbruecklii* ssp. *bulgaricus PXN39*, *L. helveticus PXN45*, and *L. salivarius PXN57* in a dose of 2 × 10^9^ CFU/capsule × 4 capsules/day achieved greater improvements in depressive symptoms as assessed by the HAMD-17 and Inventory of Depressive Symptoms-Clinician rated [IDS] scores in relation to those receiving the identical placebo. The same was also true for the Hamilton Anxiety Rating Scale [HAMA] and the General Anxiety Disorder [GAD-7] scores [54].

Gawlik-Kotelnicka O et al. [2] assessed a probiotic preparation containing *Lactobacillus helveticus Rosell^®^-52* and *Bifidobacterium longum Rosell^®^-175* in a dose of 3 × 10^9^ cfu/day or a placebo over 60 days regarding their efficacy in depressive, anxiety, and stress symptoms in 116 individuals with depressive disorders in a background of abdominal obesity or metabolic syndrome. The recruited participants with a score ≥ 13 according to MADRS were already taking antidepressants at a rate of 70% and were split into treatment and placebo groups. A significant but similar reduction in MADRS score was found in both groups at the study’s conclusion, which was predefined as a change of at least two points. Additionally, the more severe the metabolic abnormalities [overweight, excessive central adipose tissue, and liver steatosis] the smaller the reduction in psychometric scores, which also occurred with the 30% not taking antidepressants [2].

Two groups of individuals with irritable bowel syndrome [*n* = 124]—categorized as having a “normal” mood [score < 16] or subthreshold depression [score ≥ 16] according to the “CESD-R’’ [Center for Epidemiologic Studies Depression Revised form] questionnaire—were randomized to receive 12 weeks of a probiotic-containing cultured milk drink containing *Lactobacillus acidophilus LA-5* and *Lactobacillus paracasei-01* [1 × 10^9^ cfu, twice a day] or a placebo, with the aim of determining their effectiveness in reducing depression symptoms [74]. A significant reduction in CESD-R score was observed in the subthreshold depression groups, either receiving probiotics or the placebo, in contrast to the “normal” mood groups, who experienced a slight increase in CESD-R scores independent of probiotic or placebo treatment [74].

One hundred and twenty adults diagnosed with MDD were blindly randomized to receive two capsules twice daily—over 8 weeks—of the commercially available NRGBiotic™ regime (Medlab Pty Ltd., Botany, NSW, Australia) or the placebo, followed by a another 8-week follow-up period. The regime contained the postbiotics *Lactobacillus acidophilus*, *Bifidobacterium bifidum*, and *Streptococcus thermophilus* [2 × 10^10^ cfu/capsule] as well as Mg Orotate 1600 mg and CoQ10 150 mg [71]. The manifestation of MDD was evaluated using the Structured Clinical Interview [SCID-5-RV] and the Diagnostic and Statistical Manual of Mental Disorders [DSM-5] while the severity of the symptoms was evaluated using the BDI-II. The individuals having taken the combined postbiotic formula were found to have a significantly lower frequency of major depressive episodes at the end of treatment [week 8], but no difference after the 8-week follow-up period [71].

*Lactobacillus plantarum PS128*, known for having psychobiotic properties, was administered by Lin SK et al. [75] to MDD patients of moderate symptom severity [HAM-D score > 14] in order to investigate its impact on depressive symptoms. Only thirty-two patients on stable antidepressant treatment for at least 1 month were further allocated to *Lactobacillus plantarum PS128* capsules [3 × 10^10^ cfu/capsule] or a placebo for 8 weeks. Both the HAM-D and the DSSS scores showed a significant decrease in both groups [*p* < 0.001], possibly due to the small number of participants [75].

In a prospective double-blind randomized trial [72], 266 patients on a course of chemotherapy following gastrointestinal surgery for cancer were allocated to receive either a probiotic formulation containing bacteria with well-recognized psychobiotic properties or a placebo, in order to evaluate the depression status changes before and after one month of treatment and two months thereafter. The psychobiotic formula consisted of four bacteria, *Bifidobacterium animalis* subsp. *lactis* LMG P-21384 [BS01] [2.50 × 10^10^ cfu/dose], *Bifidobacterium breve* DSM 16604 [BR03] [1.00 × 10^10^ cfu/dose], *Bifidobacterium longum* DSM 16603 [BL04] [8.00 × 10^9^ cfu/dose], and *Lacticaseibacillus rhamnosus* ATCC 53103 [GG] [4.50 × 10^10^ cfu/dose], and was administered twice a day, giving a total dose of 1.76 × 10^11^ cfu/day. All 266 participants were evaluated by means of the BDI-II and the HDRS questionnaires; 99 of the participants, 48 in the probiotics group and 51 in the placebo group, were found to have depression [HADRS score > 19]. After a 4-week treatment period, a highly significant reduction in depression status was prominent in the probiotics group [*p* < 0.001] and a significant increase in the placebo group was observed [*p* < 0.01]. When assessing depression status as a dichotomous variable [depressed and non-depressed, independent of the severity], the 48 individuals with depression at baseline reduced to 22 at the end of the treatment month [RR 0.18 (95% CI: 0.10–0.31)], meaning that the risk of remaining depressed after psychobiotic treatment was 18% of the risk in the placebo group [72].

Fifty participants diagnosed with MDD [HDRS-24 > 15], after a wash-out period of at least two months prior to enrolment, began taking 20 mg of Fluoxetine daily. They were then randomly administered a probiotic formula containing *L. helveticus* R0052 and *B. longum* R0175 in a total dose of [6 × 10^−9^ cfu/day for 8 weeks, or a placebo. At the end of the 8 weeks, there was a significant reduction in HDRS score in both groups; however, the probiotic group showed a median decrease in score from 24 to 7, while for the placebo group it was from 21 to 11 [73].

### 3.3. Additional Benefits

Besides the advantageous effects of probiotics on the improvement of depression status, with this being their primary action, the psychobiotic bacteria used in the reviewed papers were shown to have additional positive actions. Five out of the nineteen articles analyzed had a total of six supplementary publications extracted from the same participants and the same study protocol and presented a post hoc analysis or similar. These additional positive findings derived from the post hoc analyses are presented—Table 3.

Both *Lactiplantibacillus plantarum 299v* [59] and *Lactiplantibacillus plantarum HEAL9* [68] have led to a significant improvement in cognitive functions, as assessed by the Work Speed in Attention and Perceptivity Test [APT] and the California Verbal Learning Test [CVLT] for the former and for the latter in cognitive and anxiety subdomains, as well as the anxiety questionnaire of Z-SAS [Zung Self-Rating Anxiety Scale]. Recently the same research team, continuing their investigation of the biochemical mechanisms involved in the improvement of cognitive functions, found *L. plantarum 299v* to enhance the effects of serotonin reuptake inhibitor antidepressants [SSRI], improving mitochondrial function through intensifying the reduction in long-chain acyl-carnitines and restoring the biochemical balance, as evidenced by the four-fold reduction in N-acyl taurines. Furthermore, *L. plantarum* was found to increase oxidized glycerol-phosphocholine, along with sphingomyelins, L-histidine, D-valine, and p-cresol, metabolic parameters known to be associated with mitochondrial dysfunction, inflammation, oxidative stress, and disruption of the gut microbiota balance [76].

**Table 3 nutrients-17-02022-t003:** Additional benefits of psychobiotic strains used in the studies.

Improvements	Psychobiotic Strains
Cognitive function	*L. plantarum 299v* [59] *L. plantarum HEAL9* [68] ***Vivomixx^®^* regime**: *Streptococcus thermophilus NCIMB 30438*, *B. breve NCIMB 30441*, *B. longum NCIMB 30435*, *B. infantis NCIMB 30436*, *L. acidophilus NCIMB 30442*, *L. plantarum NCIMB 30437*, *L. paracasei NCIMB 30439*, *L. delbrueckii* susp. *Bulgaricus NCIMB 30440* [53]
Anxiety status	*L. plantarum HEAL9* [68]
Emotional status	***Vivomixx^®^* regime**: *Streptococcus thermophilus NCIMB 30438*, *B. breve NCIMB 30441*, *B. longum NCIMB 30435*, *B. infantis NCIMB 30436*, *L. acidophilus NCIMB 30442*, *L. plantarum NCIMB 30437*, *L. paracasei NCIMB 30439*, *L. delbrueckii* susp. *Bulgaricus NCIMB 30440* [77]
Affect brain structure and function	***Vivomixx^®^* regime**: *Streptococcus thermophilus NCIMB 30438*, *B. breve NCIMB 30441*, *B. longum NCIMB 30435*, *B. infantis NCIMB 30436*, *L. acidophilus NCIMB 30442*, *L. plantarum NCIMB 30437*, *L. paracasei NCIMB 30439*, *L. delbrueckii* susp. *Bulgaricus NCIMB 30440* [78]
Increase in 5-HT and metabolites	*B. Breve CCFM1025* [27,79] *L. acidophilus LA-5* plus *L. paracasei-01* [74]
Enhance SSRI effects	*L. plantarum 299v* [76] *B. Breve CCFM1025* [27,79]
Regulation of appetite [ghrelin, leptin]	*L. helveticus R0052* plus *B. longum R0175* [80] ***Vivomixx^®^* regime**: *Streptococcus thermophilus NCIMB 30438*, *B. breve NCIMB 30441*, *B. longum NCIMB 30435*, *B. infantis NCIMB 30436*, *L. acidophilus NCIMB 30442*, *L. plantarum NCIMB 30437*, *L. paracasei NCIMB 30439*, *L. delbrueckii* susp. *Bulgaricus NCIMB 30440* [81]
Enhance immune activity	***Vivomixx^®^* regime**: *Streptococcus thermophilus NCIMB 30438*, *B. breve NCIMB 30441*, *B. longum NCIMB 30435*, *B. infantis NCIMB 30436*, *L. acidophilus NCIMB 30442*, *L. plantarum NCIMB 30437*, *L. paracasei NCIMB 30439*, *L. delbrueckii* susp. *Bulgaricus NCIMB 30440* [81]

Schaub et al. [53] worked on an eight-strain probiotic regime [*Streptococcus thermophilus NCIMB 30438*, *B. breve NCIMB 30441*, *B. longum NCIMB 30435* (re-classified as *B. lactis)*, *B. infantis NCIMB 30436* (re-classified as *B. lactis)*, *L. acidophilus NCIMB 30442*, *L. plantarum NCIMB 30437*, *L. paracasei NCIMB 30439*, and *L. delbrueckii* subsp. *Bulgaricus NCIMB 30440* (re-classified as *L. helveticus)*] and also analyzed their effects on cognitive symptoms. The participants assessed by means of the verbal learning and memory test [VLMT] revealed significantly improved immediate recall at the termination of the treatment, as well as a trend for a time and group interaction considering all time points, reflecting a highly significant improvement in the probiotic group; although both the Corsi Block Tapping Test and the Trail Making Test A and B tests failed to support the improvement. Furthermore, this probiotic formula was found to have beneficial effects on facial emotion processing, since there was a significant decrease in activation in the right and left putamen after probiotic administration, in relation to a placebo [77]. Additionally, the same research team in post hoc analysis found that this eight-strain probiotic regime affected brain structure and function in the fronto-limbic network, a result associated with a decrease in depressive symptoms [78].

Regarding kynurenine/tryptophan metabolism, when the combined formula of *Lactobacillus helveticus R0052* and *Bifidobacterium longum R0175* [*CNCM strain I-3470*] was taken, no difference was found in serum kynurenine/tryptophan ratio or tryptophan/BCAAs ratio, except when the values were adjusted to serum isoleucine. The same was found for the tryptophan/isoleucine ratio [62]. On the other hand, the subjective assessment of depression after *Bifidobacterium breve CCFM1025* treatment was supported by the laboratory findings of a significant increase in 5-HT in the probiotic group. Furthermore, the significant reduction in the serum serotonin turnover—probably related to the alterations in gut microbiome and gut tryptophan metabolism—against indole-3-acetamide which emerged in the placebo group might have affected the patients’ emotional and gastrointestinal troubles reflected in the BPRS and GSRS scores [27,79]. Additionally, eight of the tryptophan metabolites—including tryptophan, 5-hydroxytryptophan [5-HTP], 5-HT, 5-HIAA, indole-3-acetamide, and indole-3-lactic acid—were found to be up-regulated by the *B. breve CCFM1025* treatment but down-regulated in the placebo group [27]. Furthermore, when *Lactobacillus acidophilus LA-5* and *Lactobacillus paracasei-01* were taken by participants with subthreshold depression due to irritable bowel syndrome, they exhibited a significant increase in serotonin serum levels, but only a tendency toward increasing cortisol serum levels. Although, both the probiotic and the placebo groups reported an almost similar improvement in quality of life, as well as in gastrointestinal symptom severity [74].

The combined use of the postbiotics *Lactobacillus helveticus R0052* and *Bifidobacterium longum R0175* were found over time to increase the desire to eat in participants with depression, leading to a significant augmentation in energy intake and leptin, as well as a trend in hunger increase in relation to a placebo. These events seem to be related to the improvement in depression status, which in turn leads to an increase in leptin levels as a compensatory mechanism [80]. However, Schaub AC et al. [53] extended their study in treating depression using an eight-strain formula [*Streptococcus thermophilus NCIMB 30438*, *B. breve NCIMB 30441*, *B. longum NCIMB 30435* (re-classified as *B. lactis)*, *B. infantis NCIMB 30436* (re-classified as *B. lactis*), *L. acidophilus NCIMB 30442*, *L. plantarum NCIMB 30437*, *L. paracasei NCIMB 30439*, and *L. delbrueckii* subsp. *Bulgaricus NCIMB 30440* (re-classified as *L. helveticus*)], and investigated the immune inflammatory markers and the gut-related hormones ghrelin and leptin. These probiotics were found to cause a significant, though transient, increase in circulating ghrelin only during treatment, with the increase associated with the improvement in depressive symptoms [81]. Furthermore, when they performed a transcriptomic analysis, it was found that 51 up- and 57 down-regulated genes, involved in these functional pathways, enhanced immune activity. Specifically, the probiotics-dependent up-regulation of the genes ELANE, DEFA4, and OLFM4 were associated with immune activation and ghrelin concentration. The findings underline the potential of this probiotic supplementation to produce biologically meaningful changes in immune activation in patients with depression, although these were only evident during the intervention period and were no longer present at follow-up [81].

### 3.4. Gut Microbiome Changes and Depression Amelioration

Another serious issue is that of the altered gut microbiome in individuals with depression [82,83] and the ultimate goal of researchers to promote the development of microbiota-based interventions for depression. Psychobiota may be the future—Table 4.

When the OMNi-BiOTiC^®^ Stress Repair formula containing the psychobiotics *B. bifidum W23*, *B. lactis W51*, *B. lactis W52*, *L. acidophilus W22*, *L. casei W56*, *L. paracasei W20*, *L. plantarum W62*, *L. salivarius W24, and L. lactis W19* was taken by 82 individuals with depression for 28 days, a significant difference was found in the beta-diversity indices. The study found a significant increase in the *Ruminococcus gauvreauii* group in terms of global differential abundance, as well as in the taxonomically related *Coprococcus* 3 in the probiotics group in comparison to the placebo group, with *Coprococcus*—a family member of the butyrate producers Lachnospiraceae—being depleted in depression. However, no change was evident in the alpha-diversity indices at the end of the study [33].

The findings were similar when the Ecologic^®^Barrier formula, which consists of nine bacterial strains, *B. bifidum W23*, *B. lactis W51*, *B. lactis W52*, *L. acidophilus W37*, *L. brevis W63*, *L. casei W56*, *L. salivarius W24*, *L. lactis W19,* and *L. lactis W58,* was administered. There were no significant differences in the relative abundance between groups at any taxonomic level, no alteration in alpha- or beta-diversity, and no difference in the *Lactobacillus* and *Bifidobacteria* genus, being the treatment taxa. However, when the relative abundances of the operational taxonomic units and bacterial taxa were compared with the psychometric scores applied to the participants, only the *Ruminococcus gnavus* exhibited a significant positive correlation to the DASS-21 score, being present in 72% of the depressed and only in 25% of the non-depressed participants. Additionally, a higher relative abundance was identified in the severely depressed BDI-II participants, according to the BDI-II score, compared with the mild to severely depressed participants in the depressed groups [55].

Nikolova et al. [54], after having successfully treated individuals with depression with an 14-strain formula [*L. paracasei PXN37*, *L. plantarum PXN47*, *L. rhamnosus PXN54*, *B. subtilis PXN21*, *B. bifidum PXN23*, *B. breve PXN25*, *B. longum PXN30*, *L. helveticus PXN35*, *L. lactis* ssp. *lactis PXN63*, *S. thermophilus PXN66*, *B. infantis PXN27*, *L. delbruecklii* ssp. *bulgaricus PXN39*, *L. helveticus PXN45*, and *L. salivarius PXN57*], attempted to analyze the mechanisms of success. After examining stool and blood samples, they concluded that this probiotic regime did not involve a major reorganization of the gut microbiota, but only a subtle and transient, although detectable, reorganization in the overall structure [alpha- and beta-diversity]. Regarding the relative abundance of gut microbiota between groups, there was a significant decrease in Proteobacteria at 4 weeks of treatment in the probiotics group, and of Firmicutes at 8 weeks due to their decrease in the placebo group. A moderate negative correlation between *Bacillaceae*, specifically the genus *Bacillus,* and the HAM-A anxiety scores and between *Lachnospiraceae* and the IDS at the end of the treatment period also became clear. However, there were no significant differences in the BDNF, CRP, TNFa, IL-1b, IL-6, and IL-17 parameters and the depression or anxiety scores in either group [84].

When *Bifidobacterium breve CCFM1025* was administered, there were no significant changes in the beta-diversity and the core microbes; however, it is of interest that a positive correlation exists between *Blautia*, *Bifidobacterium*, and the *Lachnospiraceae FCS020* abundance and the improvement in psychometric and gastrointestinal scores in individuals with depression. Furthermore, a significant increase in the abundance of *Bifidobacterium* in the CCFM1025 group was partly attributed to the consumption of this psychobiotic [27]. Another single-strain probiotic, *L. plantarum PS128*, revealed no significant alterations in alpha- and beta-diversity, and no significant changes in gut microbiota diversity between the treatment and placebo groups. Despite these, the *Firmicutes/Bacteroid* ratio, expressing the dysbiosis score, revealed that this psychobiotic, after 8 weeks of treatment, achieved a greater average score change of 151 plus, compared with the placebo group, with a reduction of 16 [75].

Finally, the eight-probiotic formula *Streptococcus thermophilus NCIMB 30438*, *Bifidobacterium breve NCIMB 30441*, *Bifidobacterium longum NCIMB 30435* (re-classified as *B. lactis*), *Bifidobacterium infantis NCIMB 30436* (re-classified as *B. lactis*), *Lactobacillus acidophilus NCIMB 30442*, *Lactobacillus plantarum NCIMB 30437*, *Lactobacillus paracasei NCIMB 30439*, and *Lactobacillus delbrueckii* subsp. *Bulgaricus NCIMB 30440* (re-classified as *L. helveticus*) used by Schaub, A.C. et al. [53] was found to maintain α-diversity and increase the abundance of the genus *Lactobacillus*, thus indicating the effectiveness of this formula in increasing specific taxa. This increase is also correlated with the decrease in depressive symptoms, further assessed by BDI, as well as an improvement in GSRS.

## 4. Discussion

The present review is based on RCTs on adults with a well-established diagnosis of depression—based on at least one psychometric test—the severity of which is clearly stated. The participants must have received an explicitly defined probiotic treatment, in respect to the dose and treatment duration, either along with the already-taken antidepressants or alone. In addition, all studies included had to have as their primary purpose the investigation of the effects of probiotics under the study in the attenuation of depression symptoms, in a measurable manner.

Nineteen RCTs were finally deemed suitable to be included; fourteen of them included participants with depression, with no other known co-morbidity, while five studies included individuals with depression, all with the same co-morbidity or disease status—coronary artery disease [70], metabolic syndrome/abdominal obesity [2], irritable bowel syndrome [67,71], and surgical oncology patients on a chemotherapy course [72].

Fourteen out of the nineteen trials (Table 1) exhibited positive results regarding the amelioration of depression status, as assessed by the use of at least one validated psychometric questionnaire. All of these studies were based on probiotics or postbiotics considered to have well-known properties, after being tested in multiple previous trials published in high impact factor journals. In all 19 reviewed studies, probiotics were administered in adequately high concentrations, according to the present thinking [85,86], while a considerable number of them are already clearly marked as psychobiotics [87]. The remaining five studies showed negative results (Table 2), although, at first glance, they share the same technical characteristics as the positive ones. In an attempt to explain the reasons behind the failure of these studies, we analyze and comment on some specific parameters.

Four studies [2,62,69,73] were based on the same combination of two probiotics well characterized as psychobiotics, *L. helveticus Rosell^®^-52* and *B. longum Rosell^®^-175*. The first three trials with positive results included participants with depression [62,69,73], the last, however [2] included a totally non-homogenous group of rather young adults [34.4 ± 13.5 years of age], with the majority [84%] being females. In this study [2], all participants had metabolic syndrome/abdominal obesity of differing severity, which led to depression. The main primary inclusion criterion was the ICD-11 diagnosis number, that is, 6A73 for mixed depressive and anxiety disorder, representing 64.2% of participants, 6A71 for recurrent depression [rate 27.4%], and 6A70 for a depressive episode, at a rate of 8.4%. However, it should be underlined that when the participants were allocated to probiotic and placebo groups, the rate for 6A73 was 54.9% vs. 75%, for 6A71 was 31.4% vs. 22.7%, and for 6A70 was 13.7% vs. 2.3%, respectively. Furthermore, 70% of participants were taking psychotropic medication—similarly distributed in the two groups—while 27.3% and 41.2% of the probiotic and placebo groups, respectively, were additionally taking other non-psychotropic pharmacological treatment [2]. Although the difference in rates does not reach a statistically significant level, it is indicative of the population heterogeneity. Additionally, the dose used in the study was lower [3 × 10^9^ cfu/d] in relation to two other studies with positive results, that of Kazemi et al. [80] with 1 × 10^10^ cfu/d and of Elahinejad et al. [73] with 6 × 10^9^ cfu/d, but similar to that used in the study of Ullah et al. [69], who additionally treated their participants with depression with S-adenosyl methionine [SAMe], vitamin B6, and magnesium oxide, which are all considered to have antidepressant properties [88,89,90,91].

Today, there is a strong belief that prescribing a combination of probiotics is more effective than a single strain, since, in general, most probiotics act synergistically, although with different mechanisms of action [85,92]. However, each rule has its exceptions: in the present review, there are 6 trials out of the 19 which are based on only a single, well-known, probiotic strain having psychobiotic properties. In detail, the psychobiotic *L. plantarum HEAL9* along with S-adenosyl methionine [SAMe] [68], *L. rhamnosus GG* given either alone or with the prebiotic inulin [70], and the psychobiotics *B. longum* NCC3001 [67] and *B. breve CCFM1025* [27] resulted in the amelioration of depression status. However, *L. plantarum PS128* [75] and L. *plantarum 299v* [59] (both well-known psychobiotics [76,93,94]) failed to present a significant superiority against the placebo; albeit both have improved the degree of depression. In both studies, the participants were given the psychobiotic *L. plantarum* as an add-on treatment, in a higher dose of 3 × 10^10^ cfu/d [75] and 2 × 10^10^ cfu/d [59] than the four positive studies not exceeding 1 × 10^10^ cfu/d [27,67]. These findings are somewhat surprising, since *L. plantarum 299v*, given as a supplementary to antidepressant treatment exhibited positive results in other related parameters in this study [59]: an improvement in cognitive functions along with a decreased kynurenine concentration and an increase in 3-hydroxy-kynurenine to kynurenine ratio, which could be considered contributing factors to cognitive improvement [63,95]. Thus, in other words, we tentatively suggest that the non-significantly better response than the placebo is due to other, as yet unknown parameters rather than the single species, different strains, or the dose, and possibly to the number of participants and their homogeneity. Indeed, the authors themselves [75] recognize that the failure of *L. plantarum PS128* to achieve a disease remission rate, pre-defined as a 50% reduction in HAMD-17 score, is due to the small sample size [32 participants only, due to the COVID pandemic], although 26 subjects were needed in each arm for an 80% statistical power to be achieved.

A similar, unexplained, discrepancy is met in two papers testing the same nine-strain formula, commercially available as Ecologic^®^ Barrier [55] and OMNi-BiOTiC^®^ Stress Repair [33], which presented positive and negative results, respectively, despite the high attrition rate in the study population in the former. However, a recent study [96] which also used the OMNi-BiOTiC^®^ Stress Repair formula exhibited positive results; the aim being the assessment of the vagus nerve function in patients with major depression and in control patients without depression. After 3 months of treatment, the participants with depression taking probiotics demonstrated significantly improved morning vagus nerve function in relation to the controls. Furthermore, the probiotics-treated participants exhibited a significant increase in *Christensellales*, particularly *Akkermansia muciniphila*, along with improved sleep parameters. These formulas, Ecologic^®^ Barrier and the OMNi-BiOTiC^®^ Stress Repair, had six identical strains out of nine: four psychobiotics and two probiotics. The common strain was *L. acidophilus* but the differing strains were W37, being a psychobiotic, and W22, respectively; the other two different strains were the psychobiotics *L. brevis W63* and *Lc. lactis W58* for the former, and *L. paracasei W20* and *L. plantarum W62* for the latter. That means that the Ecologic^®^ Barrier included seven and the OMNi-BiOTiC^®^ Stress Repair four psychobiotic strains. The total dose was similar in the two studies, as well as the number and age of participants with depression, but the treatment duration was half [4 weeks] of that exhibiting positive effects. It is of interest to mention that in the later study [33], exhibiting negative results, the probiotics were given as an add-on treatment to antidepressants. Despite this combination, which theoretically increases the effectiveness of the treatment, the answer to the mystery of almost significantly equal improvements in depression status seems to come from two other interventions: (i) both groups were taking supplementary treatments with 125 mg D-Biotin [vitamin B7], common horsetail, fish collagen, and keratin plus matrix, which, in our opinion significantly ameliorated the depression status of both probiotic and placebo groups; (ii) all subjects recruited to the study were inpatients of the Department of Psychiatry—as also occurred with the patients participating in the study performed by Schaub et al. [53]—and were receiving additional treatments, such as occupational therapy and psychotherapy, throughout the study period. Furthermore, the authors [33] attributed the failure to the disturbance of the participants’ nutritional habits, since, being inpatients, they had to mandatorily receive hospital meals only. The baseline differences in smoking frequency can be considered to have a confounding influence on the results, as well as the higher number of female participants.

The final study exhibiting negative results was that of Sarkawi et al. [74]. They tested a cultured milk drink containing *L. acidophilus LA-5* and *L. paracasei-01* for 12 weeks in patients with IBS fulfilling the IV Rome criteria, who were either experiencing subthreshold depression [CESD-R ≥ 16] or having non-depressive moods [CESD-R < 16]. The participants were assigned to one of four groups, that is 27 to 28 subjects per study group. Regarding participants’ homogeneity, 40% were assigned to the IBS subgroup with diarrhea as the predominant symptom and 30% as IBS–constipation; however, there is no data for the four groups separately. This disadvantage, besides the obvious question of the ratio within each group, also raises the concern of the depression severity in each symptom-related IBS category. The authors [74] consider as the obvious limitations of their study: the unassessed nutritional delivery and physical activity of the participants, along with the implementation of the low FODMAP diet some of them received [89,97]; furthermore, the presence of fructose and polydextrose as ingredients of the cultured milk drink used could potentially affect gut motility and, thus, psychosomatic functions. Finally, another question remains as to whether there was a change in depressed participants to a non-depressed state at the end of the treatment time or at least a change from a high or median rank of the clinical diagnosis to a low rank or to a sub-clinical diagnosis.

Two studies attempt to answer this. Besides the reduction in the depression score in absolute numbers, numerical data on the movement of participants from very severe to severe or moderate or, finally, to a non-depressed depression score, according to the HDRS or similar scores, was given and analyzed in a four-strain psychobiotic study [*B. animalis* subsp. *Lactis LMG P-21384*, *B. breve DSM 16604*, *B. longum DSM 16603*, and *L. rhamnosus ATCC 53103*] on gastrointestinal cancer patients on a post-surgery chemotherapy course [72]. In this study, the participants were divided into one of four groups based on depressed or non-depressed scores after having been randomly assigned to psychobiotic or placebo groups. It is of interest to see the significant improvement in the patients to a less severe depression status or even to a non-depressed status when taking psychobiotics, while, in parallel, there is a dramatic downturn of non-depressed participants receiving placebo into a depression status over time; this directly related to the inability of their emotional burden to cope with the recrudescence of their underlying disease status, that is gastrointestinal cancer and chemotherapy. In such a patient cohort, there is an inherent tendency for participants to slide towards deeper depression over time, whether receiving probiotics or not, since, despite chemotherapy, patients either develop complications or simply lose weight and muscle strength, thus feeling the end approaching. To make it easier to understand, we simply state that at the end of the month of treatment, the 48 patients with depression decreased to only 22, while the patients without depression increased from 51 to 71. A similar effort was made by Chahwan et al. [55] to assess the changes in the percentage of patients classified according to the depression score before treatment; unfortunately, although a down-regulation trend exists, the number of participants in each subgroup is too small to draw significant differences—i.e., 14 patients being in the “clinical” subgroup according to M.I.N.I. classification decreased to 10 after probiotic treatment.

Another topic which needs to be discussed is subthreshold depression. Subthreshold depression (also referred to as subclinical depression, although there is no universally accepted definition), is a form of subclinical depression with an increased risk of major depressive disorder [74,98]. It is defined by at least two depressive symptoms manifesting for at least two weeks, but without reaching the minimum diagnostic criteria for dysthymia and/or major depression [69,99]. Furthermore, these patients may not be eligible for standard antidepressant treatment, such as tricyclic antidepressants [TCAs] or selective serotonin reuptake inhibitors [SSRIs] or, when given, the treatments may lead to adverse effects which outweigh their therapeutic potentials, leading to therapy discontinuation [69]; thus, their use in subthreshold depression is still controversial and the overall therapeutic strategies remain mostly underdefined [68,100]. Additionally, according to a meta-analysis of these antidepressants’ effects on depression severity, Fournier et al. [101] conclude that the degree of effectiveness of these drugs is strictly associated with the baseline severity of symptoms; the more severe the depression, the more significant the improvements, while minimal changes are observed in those with subthreshold, mild-to-moderate levels. In our review, there are three studies concerning participants with subthreshold depression [68,69,74]. In two [68,69] out of these three studies [68,69,74], the psychobiotics given as an add-on treatment to the SAMe regime exhibited a significant improvement in depression status over the placebo. In the third study [74], in IBS patients with subthreshold depression—identified as frequent in 32.1% of IBS patients [102]—the combined formula of *L. acidophilus LA-5* and *L. paracasei-01* in cultured milk failed to achieve a difference in the depression status over the placebo group, which also presented an improvement in symptoms, with the authors attributing the findings to the different IBS subtypes, among other factors.

In the last few years, it seems that another practice is emerging, that of high dose and short-term duration. The first attempt was made with the study of Reininghaus et al. [33], who tested the OMNi-BiOTiC^®^ Stress Repair, a nine-strain probiotic regime, taken as an add-on for a 4-week treatment period, but in a dose of 7.5 × 10^9^ cfu/d—not to be considered a low dose in the era of 2020. However, the results did not support the aspiration of the authors. Schaub et al. [53] tried with another commercially available eight-strain regime, the Vivomixx^®^ (Mendes SA, Lugano, Switzerland), as an add-on therapy for 4 weeks at a very high dose of 9 × 10^11^ cfu/d, with their results confirming the adequacy of the dose and length of time, as well as the effectiveness of the probiotic strains in ameliorating depression along with psychobiotics. Tzikos et al. [72] used the four-strain psychobiotic formula of the commercially available Lacto-Levure Probio-Mood^®^ (Uni-Pharma, SA, Athens, Greece), after removing the add-on magnesium, FOS, and saffron extract, at a dose of 1.76 × 10^11^ cfu/d for 4 weeks in patients on a chemotherapy course after gastrointestinal cancer surgery. Although such patients may be on a continuous trajectory towards depression due to the progressive deterioration of their disease, despite various treatments, the results are most promising. The concept of short-term treatment certainly serves the rapidly deteriorating psychiatric status of the participants in the study; however, it also somehow protects the researcher from high attrition rates due to treatment discontinuation, a phenomenon more common in the psychiatric population than in controls without depression. Regarding the high dose, it is dictated as an effort to boost the gut–brain axis towards better collaboration, also keeping in mind the as yet unknown time period that the probiotics remain in the gut, since many studies with encouraging results failed to record results at follow-up.

Finally, it would be of interest to discuss the option of taking psychobiotics as an add-on treatment for patients with depression, whether this would have essential importance in the outcome or not. To date, to the best of our knowledge, there is only sparse data on the possible mechanisms by which psychobiotics are involved in the enhancement of the effectiveness of antidepressants. In a recent in-depth investigation, Godzien, J. et al. [76] attempted to elucidate the biochemical mechanisms underlying depression and cognitive disfunction, as well as the role of selective serotonin reuptake inhibitor [SSRIs] treatment alone or with the psychobiotic strain *Lactobacillus plantarum 299v*. They documented that SSRI administration affects the acyl-carnitine levels, mainly by reducing the medium- and long-chain acyl-carnitines, which are related to mitochondrial dysfunction noted in individuals with depression. It may restore mitochondrial β-oxidation, thus promoting the use of medium- and long-chain acyl-carnitines, while *L. plantarum 299v* co-treatment further improves this action. Furthermore, the existence of increased taurine levels, being the primary product of N-acyl taurine degradation, is well known from the distant past to be closely related to depression severity [103], with antidepressant treatment designed to reduce them [104], while *L. plantarum 299v* administration—against placebo treatment—achieves a fourfold greater reduction in N-acyl taurines. Furthermore, probiotics received as an adjunctive treatment in a previously analyzed study [53] were found to prevent neuronal degeneration along the uncinate fasciculus and to alter the fronto-limbic resting state functional connectivity in participants experiencing an improvement in depression symptoms [78]. Additionally, Rayan et al. [105] examine the transcriptomic effects of three classes of antidepressants along with a psychobiotic regime containing *L. rhamnosus* Rosell^®^-11 and *L. helveticus* Rosell^®^-52 on 10 regions of the human and mouse brain. They reported that the treatment leads to a complex antidepressant plus probiotic action on the brain and identified pharmacological-specific cellular processes as well as gene targets related to depression.

In the present review, of the 19 studies analyzed, 10 of them administered probiotics as an adjuvant treatment; 8 studies administered probiotics along with antidepressants [33,53,54,59,61,73,75,80]; and another 2 administered probiotics with SAMe [S-Adenosyl Methionine] [68,69], considered a naturally occurring psychobiotic, almost routinely subscribed in Canada [106] as an antidepressant treatment [89]. In another 5 studies [55,67,70,72,74], it was clearly stated that individuals taking antidepressants were excluded from allocation; all but 1 [55] of these studies dealt with depression related to IBS [67,74], coronary artery disease [70], and gastrointestinal cancer with chemotherapy [72]. Finally, there were another 4 studies, in which there was no restriction on participation regarding whether the person was receiving antidepressants or not: in 2 studies, 90% [27] and 70% [2] of participants were on such a treatment; in another study [71] almost half the participants [40%] were taking antidepressants, and in the last study [42] there was no relevant report. On the other hand, seven out of the nine studies [rate 77%] which included individuals without any specific criterion [2,27,42,55,67,70,71,72,74] relating to participants taking antidepressants also revealed positive results [27,42,55,67,70,71,72], while the remaining two [2,74] referred to patients with depression also with metabolic syndrome and IBS, respectively. Our results could thus have a double reading. First, that psychobiotics given as an add-on treatment show positive results in enhancing the antidepressants’ effect in 7 out of the 10 studies, the success rate being 70% [53,54,61,68,69,73,80]. On the other hand, 7 out of the remaining 9 studies [rate 77%] [27,42,55,67,70,71,72], where participants entered the study without limitations on the taking or otherwise of antidepressants, also exhibited positive results. Although the numbers are too small to extract reliable conclusions, we might suggest that the psychotropic effectiveness of the treatments are related to the probiotic strain(s) rather than to the antidepressant itself. In other words, we suggest that a strong effect psychobiotic regime has no need for an antidepressant, as is possibly necessary for a less effective psychobiotic, either due to the strain or the dose. We should also underline the high ineffective rate of antidepressants; according to Barlati et al. [14] almost one-third of patients entering a new antidepressant regime may have treatment-resistant depression, a rate equal to that achieving remission. Given that almost every study has tested a different probiotic regime [single or multi-strain]—15 regimes for 19 studies—the different doses used and the rather non-homogeneous group of participants, with respect to disease severity, as well as to their age, genus, use/lack of use of antidepressants, and overall small number per group, makes it difficult to discriminate between effective and less effective regimes, which seem to be another parameter affecting the results and leading to treatment failure.

**In conclusion**, all of the above findings lead us to modestly support a ‘trial-and-error’ approach, until an effective therapy is found. When a patient is under antidepressant treatment, psychobiotics can act as a complement: if they are not taking antidepressants, they can take psychobiotics alone and be monitored for one to three months; when there is a positive response, they can continue with them, or, otherwise, an alternative better tested regime can be tried, or antidepressants can be added, one step at a time. Furthermore, we should once more consider and strongly suggest the option of high doses and short-term treatment, in combination with the aforementioned advice to ‘add and monitor’ for new-entry participants, since, certainly, there is a difference between an RCT and real life. 

## 5. Final Thoughts and Future Perspectives

In 2014, just a year after the first publication on probiotics which exert psychotropic effects, named “psychobiotics” by Dinan et al. [20], the International Scientific Association for Probiotics and Prebiotics [ISAPP], in their consensus meeting on the mechanisms underlying probiotic effects, suggested three possible mechanisms of action: the “widespread”, the “frequent”, and the “rare”. Under the rare mechanisms, they included the newly recognized “gut–brain axis” and the involvement of pro[psycho]biotics in it, in a strictly strain-specific action [85]. Ten years later, there are more than a hundred distinct probiotic strains, well characterized as psychobiotics and all exerting diverse, strain-specific properties. Despite that fact that there is a widespread belief that psychobiotics exert beneficial effects on different mood disorders, there is unfortunately no consensus within different RCTs, and, thus, systematic reviews cannot make any conclusions with any certainty, while acknowledging the high heterogeneity and poor evidence in many of them. The same problem occurs with probiotics, given for a lot of diseases-different populations, along with low doses, and absence of statistical power. However, as time has passed, probiotics have become widely established in the minds of both “patients” and physicians. The same will become reality with psychobiotics, too. Since the rhythms of life in the Western world lead and have led to a dramatically increasing number of mood disorders and the use of antidepressants does not fit all the needs or expectations to any great degree for those who take them and many have significant side effects, we believe that psychobiotics will become established in a much shorter period of time. However, it must be emphasized that there is an urgent need for large-scale randomized clinical trials with well-defined specific strains in order for the meta-analysis of studies using the same strains to be possible. Only then will there be statistical confirmation of their effectiveness.

## Figures and Tables

**Figure 1 nutrients-17-02022-f001:**
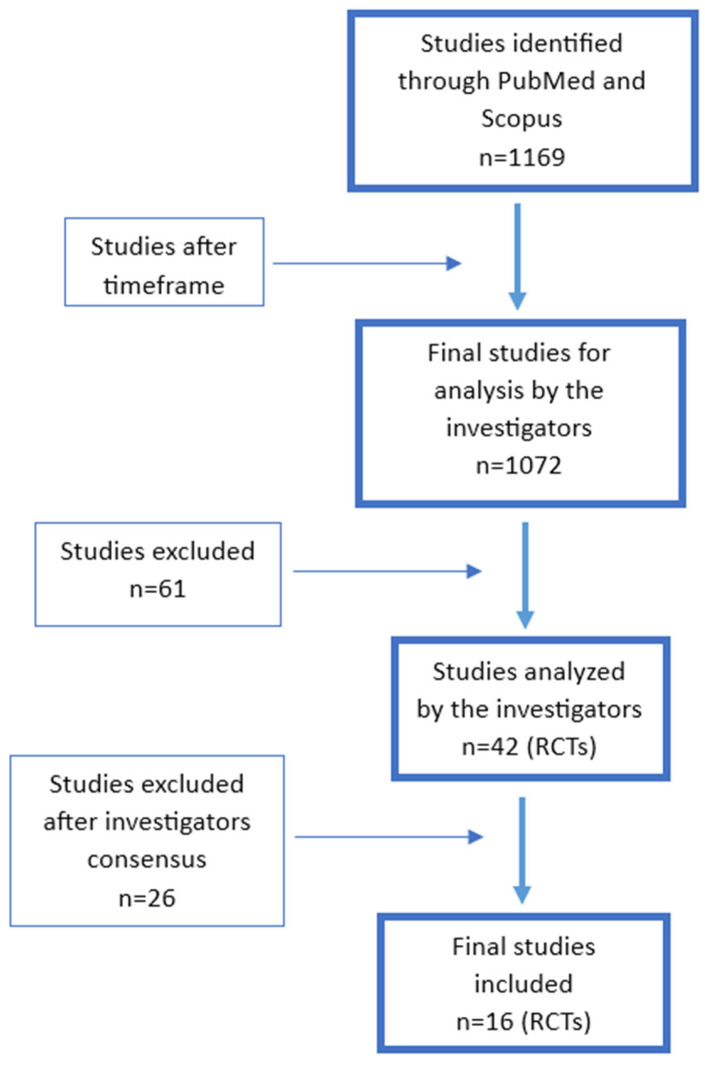
Flowchart.

**Table 4 nutrients-17-02022-t004:** Psychobiotics-induced gut microbiome changes.

Psychobiotic Strains	Gut Microbiome Changes
*OMNi-BiOTiC^®^ Stress Repair: B. bifidum W23*, *B. lactis W51*, *B. lactis W52*, *L. acidophilus W22*, *L. casei W56*, *L. paracasei W20*, *L. plantarum W62*, *L. salivarius W24, L. lactis W19*	increase in the *Ruminococcus gauvreauii* group increase in *Coprococcus* 3, a family member of the butyrate producers *Lachnospiraceae* [33]
**Ecologic^®^ Barrier:** *B. bifidum W23*, *B. lactis W51*, *B. lactis W52*, *L. acidophilus W37*, *L. brevis W63*, *L. casei W56*, *L. salivarius W24*, *Lc. lactis W19*, *Lc. lactis W58*	decrease in Ruminococcus gnavus [55]
**BioKult Advanced:** *L. paracasei PXN37*, *L. plantarum PXN47*, *L. rhamnosus PXN54*, *B. subtilis PXN21*, *B. bifidum PXN23*, *B. breve PXN25*, *B. longum PXN30*, *L. helveticus PXN35*, *L. lactis* ssp. *lactis PXN63*, *S. thermophilus PXN66*, *B. infantis PXN27*, *L. delbruecklii* ssp. *bulgaricus PXN39*, *L. helveticus PXN45*, *L. salivarius PXN57*	decrease in *Proteobacteria* [after 4 weeks] and *Firmicutes* [after 8 weeks treatment] [84] increase in Lachnospiraceae [85] increase in *Bacillaceae*, and specifically in genus *Bacillus* [84]
Bifidobacterium breve CCFM1025	increase in *Blautia*, *Bifidobacterium*, and *Lachnospiraceae FCS020* [27]
*L. plantarum PS128*	improvement in *Firmicutes/Bacteroid* ratio [75]
**Vivomixx^®^:** *Streptococcus thermophilus NCIMB 30438*, *B. breve NCIMB 30441*, *B. longum NCIMB 30435*, *B. infantis NCIMB 30436*, *L. acidophilus NCIMB 30442*, *L. plantarum NCIMB 30437*, *L. paracasei NCIMB 30439*, *L. delbrueckii* susp. *Bulgaricus NCIMB 30440*	increase in the abundance of the genus *Lactobacillus* [53]

## Data Availability

Not applicable.

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
