# Peer review of "Rewiring Mood: Precision Psychobiotics as Adjunct or Stand-Alone Therapy in Depression Using Insights from 19 Randomized Controlled Trials in Adults"

_nutrients, 2025, doi:10.3390/nu17122022_

Round 1

Reviewer 1 Report

Comments and Suggestions for Authors

I read the manuscript entitled: “Rewiring Mood: Precision Psychobiotics as Adjunct or Stand-alone Therapy in Depression” with great interest. The authors have taken up an extremely important and current topic of the growing problem of mental disorders, including depressive disorders, on a global scale and the possibility of using monotherapy and/or adjunctive therapy – psycho(pro)biotics. The authors also summarized the potential mechanisms by which psycho(pro)biotics could alleviate depressive symptoms..

I believe that it has been prepared exhaustively and methodically correctly. Below are a few comments that I would recommend taking into account in the preparation of the final version of the manuscript. The manuscript provided for review was not prepared on a form containing line numbering, so indicating specific fragments requiring correction is difficult.

All abbreviations used in the text should be explained in the first place of their use, except for some indicated in the editorial requirements of the Journal. The authors have included a glossary of abbreviations on pages 18-19, but in accordance with the editorial requirements of the Journal, as well as for the convenience of the reader, explanations should be included in the first place where a given abbreviation is used.

It is erroneous to utilise the present tense to refer to actions undertaken by the authors of the manuscript, for example, on page 3 in the final paragraph of section 1. Introduction: The objective of the present review is to search for (...) The work has already been completed, the manuscript has been written, so the past tense should be used ("Thus, the aim of the present review was to search (...)").

A comprehensive review of the text is also necessary to identify and rectify any editorial errors.

Tables 1 and 2 should be relocated to section 3. Results, specifically to the beginning of subsection 3.2. Studies overview. Tables contain characteristics of the studies used in the review and then characterized in this subsection, so they shouldn't be placed in section 4. Discussion. Both tables need to be reformatted and adjusted to the editorial requirements of the journal. In addition, I would suggest adding two additional columns: country in which the study was conducted and diagnosis or severity (type) of depression. I believe that in this form the tables would contain a full description of the studies that the authors have included in the manuscript. Additionally, it is worth explaining/specifying before the tables what the terms used in the table titles mean: "... positive results ..." (table 1) and "... negative results ..." (table 2). And one last thing: table headings should start with a capital letter.

Items 3 and 8 from the References section require adjustment to the Journal's editorial requirements.

Reviewer 2 Report

Comments and Suggestions for Authors

The manuscript presents a critical revision of RCTs primarily focused on evaluating the efficacy of probiotics/psychobiotics in ameliorating depressive symptoms. Starting from a very large number of studies (>1000), the selection criteria narrow the analysis down to 19 papers. The manuscript is conducted rigorously and is well written, although the analysis of the results is somewhat scattered and verbose. However it is too long, and it is necessary to better focus on the most relevant issues.

The absence of  line numbers  hinders an accurate point by point revision.

Major issues:

It would be useful to include an introductory section that outlines the criteria and the various indices used for the quantitative assessment of the pathology, possibly with a table clearly indicating the and the upper limits of each scale and the threshold values.

It is important to explain why a so large number of manuscripts was excluded from the analysis.

The manuscript frequently refers to POSTBIOTICS, often indicating a CFU/g value that suggests the presence of live bacteria. The authors should either clarify what is meant by postbiotics, referencing scientific literature for the definition, or revise the text where the term is used inappropriately.

“19 articles remained for analysis, 5 of which had a total of 6 supplementary publications extracted from the same participants and the same study protocol and presented as a post-hoc analysis or similar”. A schematic representation of the features of the 19 studies is required at the beginning of the results section, merging Table 1 and Table 2. The studies should be grouped according to the probiotic composition. Taking into account the limited number of studies, they can have an ID number to make easier in the text the association to the corresponding study and avoiding the use of the reference number. The table should report the outcome of the study in terms of efficacy against the depression,  with the different scores of placebo and probiotic treatments and the evaluation scale.

In the text, the outcome of the 19 studies can be reported in a more concise form, analyzing together those that use the same probiotic formulation, highlighting the main results and the molecular mechanism evoked (now in the section Additional benefits), as well as the changes in terms of microbiota composition.

The discussion should not repeat contents already presented in the results and/or introduction. It is important to emphasize the fact that comparing results or isolating the role of the probiotic is very difficult, as in many studies the probiotic is administered alongside antidepressants, while in others it is taken together with additional supplements. This makes any comparative evaluation of the results impossible, as each study must be considered as a separate case.
